# Interpreting Emergent Features in Deep Learning-based Side-channel Analysis

**Sengim Karayalçin**
Leiden University, The Netherlands
s.karayalcin@liacs.leidenuniv.nl

**Marina Krček**
Radboud University, The Netherlands
marina.krcek@ru.nl

**Stjepan Picek**
University of Zagreb Faculty of Electrical Engineering and Computing, Croatia &
Radboud University, The Netherlands
stjepan.picek@ru.nl

## Abstract

Side-channel analysis (SCA) poses a real-world threat by exploiting unintentional physical signals to extract secret information from secure devices. Evaluation labs also use the same techniques to certify device security. In recent years, deep learning has emerged as a prominent method for SCA, achieving state-of-the-art attack performance at the cost of interpretability. Understanding how neural networks extract secrets is crucial for security evaluators aiming to defend against such attacks, as only by understanding the attack can one propose better countermeasures.

In this work, we apply mechanistic interpretability to neural networks trained for SCA, revealing *how* models exploit *what* leakage in side-channel traces. We focus on sudden jumps in performance to reverse engineer learned representations, ultimately recovering secret masks and moving the evaluation process from black-box to white-box. Our results show that mechanistic interpretability can scale to realistic SCA settings, even when relevant inputs are sparse, model accuracies are low, and side-channel protections prevent standard input interventions.

## 1 Introduction

Side-channel analysis (SCA) is a realistic security threat that consists of diverse methods that allow for the extraction and exploitation of unintentionally observable information of internally processed data [23]. SCA enables the establishment of a relationship between passively observable information and the internal state of a device under investigation. As such, it poses a major threat to devices that handle sensitive data like keys, private certificates, or intellectual property (see, e.g., [40, 41]). In SCA, sensitive information gets extracted from a device by observing its physical characteristics during computation (e.g., power consumption, timing).

Since 2016 [25], deep learning-based side-channel analysis (DLSCA) has received significant attention from the research community [36]. The main benefits of using deep learning (DL) over classical techniques are that assumptions about attacker capabilities can be relaxed, leading to better attack performance. Thus, integration of these techniques into evaluation procedures for cryptographic implementations has become standardized [12].

One of the main open challenges for black-box evaluations using DL is interpretability [36]. A model that can extract the key confirms there is some exploitable leakage, but does not indicate *how* the network exploits *what* leakage. Notably, this does not allow the evaluator to provide any feedback

39th Conference on Neural Information Processing Systems (NeurIPS 2025).

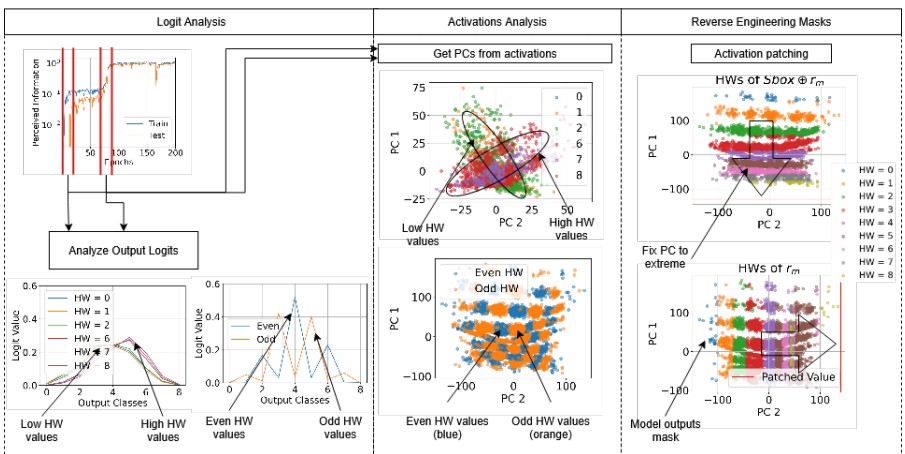

Figure 1: The analysis approach used in this study broadly consists of three major steps. After the performance increases are located using the PI metric, we plot logits to extract relevant features. Using these features, we plot the PCs of the activations and find the structure related to the leakage. Finally, we apply activation patching to reverse-engineer the masks.

beyond pass/fail, which complicates the cost-effective implementation of a solution. Understanding how neural networks learn to exploit side-channel information can prove crucial for developing robust defenses against these attacks [39]. Thus, several attempts have been made to understand network behavior. However, these approaches either focus only on input visualization [26, 18], use more explainable model architectures [52, 53], or require access to masking randomness [54, 35]. We discuss this related work in Appendix A.

Although interpreting how neural networks perform computations is generally difficult, the algorithmic tasks performed in models trained on side-channel data are conceptually relatively simple. Learning to extract leakage information from masked implementations is similar to toy models that learn group operations in the works on grokking [37, 30, 7, 56]. Concretely, for masked implementations, the computations on a sensitive value $s$ are split into $d$ secret shares $s = s_1 \cdot s_2 \cdots s_d$. Then, to learn to extract leakage from side-channel signals, a neural network needs to combine leakage from each of these shares, often without the knowledge of individual shares even for the training set [27].

The connection between side-channel and grokking models is further motivated by the observation of Masure et al. that the learning curves for models trained against masked targets show an 'initial plateau' (Section 5.2 of [27]). After a number of training steps where test loss does not improve, the models suddenly generalize to the test set and can extract the (sub)key. These sudden increases in performance raise the question of what the model is learning. Indeed, as some models for neural scaling predict neural networks learn in discrete steps [28], we expect that investigating what is learned during these transitions will give a reasonable understanding of model behavior. Recent successful results of mechanistic interpretability (MI) investigating sudden generalization[1] in toy models [30, 42] and even language models [33] further motivate this direction.

From the point of view of MI, side-channel data provides an interesting test case. The data is often noisy, high-dimensional, characterized by subtle dependencies that are difficult to capture and interpret, and presents a real-world scenario. Additionally, the masks are hidden values and should not be publicly accessible[2] which further complicates the application of MI as we cannot describe model behavior exhaustively with respect to concrete input features as in [30, 7], or do (automated) input interventions to align with a causal model as in [14, 8].

In this work, we aim to understand **what specific side-channel leakage a successful network has learned to exploit**. Concretely, we derive features from model outputs, find geometric structures that emerge in principal components (PCs) during sudden jumps in performance (phase transitions), and relate these to the physical leakage. As a practical consequence, we utilize this emergent structure to

---

[1]Sometimes referred to as phase transitions in related works [30, 33].

[2]Even in evaluation contexts with collaboration from developers, this is often impossible; see the introduction of [27] for an in-depth discussion.

extract input features, i.e., individual shares $s_i$ related to device internal randomness, from model activations, providing a path to move from black-box to white-box evaluations. The overall analysis process is illustrated in Figure 1.

To summarize, our main contributions are:

- We explore the feasibility of applying MI in a challenging real-world setting where input interventions to features are not possible due to SCA countermeasures.
- By investigating the changes in model outputs during sudden jumps in model performance, we find how networks combine leakage in DLSCA.
- We directly retrieve the internal secret share values by applying activation patches[3] to intermediate layer activations across several targets.
- We provide more detailed insights into the specific physical leakage that neural networks exploit for widely used (DL)SCA benchmark datasets. Notably, we do this without assuming a priori mask knowledge [54, 35] or requiring custom architectures [52, 53].
- We find identical structures emerging during sudden generalizations for models trained on side-channel traces captured on different implementations and in different SCA domains (electromagnetic vs. power), providing further evidence for the weak universality hypothesis [7].

Code to reproduce experiments is available at `https://github.com/Sengim/feature_emergence`.

## 2 Background

**Deep learning-based SCA (DLSCA):** The main principle behind SCA is that during the execution of an algorithm on a physical device, side-channel information, e.g., power or electromagnetic (EM) measurements, can be influenced by secret-dependent internal computation [23]. For example, under standard assumptions, writing 0000 to a register will consume less power than 1111. The goal of SCA is then to extract this secret from the side-channel observations to establish security bounds for devices that operate in conditions where physical access may be possible for attackers (e.g., bank cards, passports, mobile phones). While many SCA variants exist, a common division is into direct and profiling attacks [36]. Direct attacks assume a single device where the attacker uses (classical) statistical techniques to find the most likely key.

In profiled attacks, which are often used by device evaluation labs [6], one assumes the attacker can access a copy of the device to be attacked. This copy is under the complete control of the attacker and is used to build a model of the device. The attacker uses that model to attack a different (but similar) device. While profiled attacks are less practical due to the assumption of access to a copy of a device, it can be significantly more powerful than direct attacks. Indeed, provided that the model is well-built and there is sufficient information in the trace, one could need only a single trace from the device under attack to obtain the secret key. On the other hand, direct attacks may require significantly more traces to break a real-world target [36].One can easily observe a similarity between profiled attacks and the supervised machine learning paradigm (where building a model is training, and the attack is testing). Consequently, in the last decade and more, many machine (deep) learning algorithms have been tested in SCA [36].

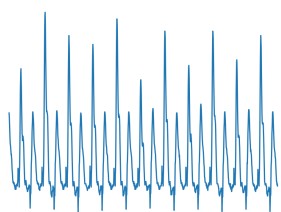

Figure 2: Example of a single trace captured during the execution of the Advanced Encryption Standard (AES) cipher [9].

The main workflow for SCA is to take a (large) number of traces (see Figure 2 for an example) from the profiling device with known key(s). These traces, often containing anywhere from 100-10 000 points per trace[4], are then labeled using an intermediate value used during the computation that

---

[3]Activation patching is a technique from MI.

[4]Generally, only a small number of points in the trace contains relevant information as the intermediate value is only used during few instructions, complicating classical SCA techniques.

depends on some public input (i.e., plaintext) and some sensitive data (i.e., the secret key). The exact mapping between the trace and intermediate values also depends on the assumed leakage model. An evaluator can employ Identity (ID) leakage model, which takes the intermediate values directly, or use leakage models that assume some physical leakage function, e.g., Hamming Weight (HW) or bitwise models. The chosen leakage model then also directly influences the number of labels (e.g., when working with a byte-oriented cipher like AES, the HW leakage model results in 9 classes, while the ID leakage model results in 256 classes). For AES, the label is often the output of the S-Box in the first round (S-Box$[p \oplus k]$) since it only depends on one plaintext and key byte ($p$ and $k$). The neural network is then directly trained on these labeled traces, eliminating the need for labor-intensive (and often error-prone) feature engineering processes [36].

In the attack phase, the trained model is used to predict intermediate values from the traces of the target device. As models often have accuracies that are only marginally above random guessing, evaluating success is done by accumulating the predictions across a larger number of traces and evaluating which of the hypothetical key candidates[5] is the most likely. In evaluation settings, the model is then said to 'break' the device if the correct key is the top candidate within some specified number of traces [44]. Other information-theoretic metrics aim to directly quantify the secret information present in a single trace. Perceived Information (PI) [38], an easier-to-estimate lower bound on mutual information, is often used in DLSCA settings. Intuitively, a PI of 0.5 means there are 0.5 bits of key-related leakage in a single trace.

To protect against SCA, countermeasures such as hiding or masking are commonly used. In both cases, the goal is to remove the correlation between the observed quantity (traces) and secret information. Hiding countermeasures can happen in the amplitude domain by randomizing/smoothing the signal or by adding desynchronization/random delays in the time domain. Masking [20], on the other hand, divides a secret variable into several shares such that one needs to know all the shares to obtain the secret information. For instance, consider a Boolean masking of a secret variable $s$. If we combine that secret variable with a random value $m$, we obtain a new variable $y$: $y = s \oplus m$. Then, to obtain information about $s$, one needs to know both $y$ and $m$. DLSCA can often automatically circumvent these countermeasures and still result in extremely efficient attacks without requiring additional access assumptions (e.g., the ability to disable countermeasures on the copy device). For a practical introduction to DLSCA, we refer readers to [22], and for a broader overview of SCA, see [43].

**Mechanistic Interpretability:** Mechanistic interpretability (MI) aims to reverse engineer a neural network into human-understandable algorithms [32, 31, 48, 30]. This involves identifying "features", which are directions in internal representations that correspond to concepts, and "circuits", which are subgraphs within the network composed of interconnected neurons and their weights, representing meaningful computations. Generally, the first step in the process of MI is to identify the features. Examples of features include low-level features such as curve or edge detector neurons in vision models [32], or more high-level features corresponding to the board state in toy models trained on board games [24, 29]. As features generally correspond to linear directions in the latent space, training linear probes [1], i.e., small classifiers, is common for showing the presence of features in the latent space.
After finding features, the goal becomes to determine how these features relate to model outputs (or other features). Ideally, we can create a causal abstraction of network behavior based on feature descriptions [14]. One method for showing causal effects involves intervening in model activations by performing activation patches [17]. Here, we replace (part of the) activations during a forward pass with saved activations from another forward pass corresponding to a different feature value to understand the effects on model outputs. This allows for measuring the impact of a specific feature or, eventually, verifying that the circuit is a (faithful) description of the model behavior.

## 3 Analysis Approach

The analysis process is shown in Figure 1 and detailed in this section. However, additional analysis and MI techniques might have been used depending on the observed behavior and findings of each specific dataset. These additional steps and the reasons for them will be described directly in the experimental results (Section 4).

---

[5]As we target a byte at a time, exhaustively searching over 256 values is easy.

**Assumptions.** In (DL)SCA, the attack typically focuses on extracting a subkey (often a single key byte) of the secret key. As we target post-hoc analysis of successful models, our analysis assumes that the attack has already succeeded and the subkey has been recovered. This assumption allows us to label (test) traces by deriving the intermediate value (label) from the input (plaintext) and the key (recall that we target S-box$[p \oplus k]$). We do not assume we have access to mask values. **The goal is to understand the model's behavior and identify what information it extracts from the traces to make predictions.** Additionally, we aim to recover the masks used in the cryptographic algorithm, which enables us to recover the rest of the secret key with (significantly) less effort. Note that when the model fails to recover the correct subkey, suggesting that the underlying masks and feature representations were not properly learned, interpretability methods offer limited insight.

**Logit Analysis.** Once we have a model that successfully recovers a subkey, our primary goal during initial exploratory testing is to understand the factors influencing the network's predictions. To achieve this, we analyze models at points directly after generalization and observe changes in the model predictions. As these sudden changes in performance suggest significant changes in the neural network's behavior during training, they are shown to be useful for discerning features [56]. We examine the distributions of output logits for different classes, looking for clear separations between classes, indicating distinct patterns in the traces. We aggregate the distribution for model outputs for traces that belong to each class and visualize them to identify commonly confused classes. These insights enable us to formulate hypotheses about higher-level features influencing predictions. Opposed to other recent works that reverse engineer models, see, e.g., [30, 48], where the authors assign features to (or derive features from) model inputs, we rely on output logits as we do not have access to masking randomness. Additionally, the (physical) noise inherent to side-channel traces results in final model accuracies that can be only marginally above random guessing, making single trace predictions challenging to analyze from the MI perspective. Note that the analysis becomes easier in white-box SCA settings, where one would assume knowledge of all internal values during computation, including the masking randomness, see [35].

**Activations Analysis.** After finding and testing initial hypotheses about the physical leakage used for classification, we can look at activations and how these relate to the predictions. Considering that in SCA only a small number of operations in each trace should be relevant for the classification (i.e., only leakage related to the target value), the number of relevant features should be relatively small. This means PCA will likely reveal the features the model learns during the first phase transitions [42, 56]. Note that for more complicated tasks where there are more features than dimensions, sparse autoencoders can be an alternative to extract features [10, 4].

As we expect structure to emerge in the first few PCs [42], we can plot the distribution of attack traces for features we derive from the logit analysis. Still, while we expect a specific structure would emerge in the first few PCs, some manual effort in determining the correct (number of) components and subdivisions might be necessary. However, in the tested cases, we notice the structure generally emerges with up to the first four components. Ideally, we see clear divisions between groups of traces belonging to certain values. Even if this is not the case due to noise, some regions might contain more/less of certain groups, and the overall distribution should be tied to (noisy) physical leakage. After finding some structure, we should explain how it arises in terms of the physical leakage that is in the trace. For example, if there is a grid-like structure in the PCs, we could assume that (embeddings of) two secret shares correspond to the $x - y$ directions of the grid since we have two shares.[6]

**Reverse engineering masks with activation patching.** When a structure is found, we need to verify that the hypothesized behavior is causally related to the model predictions in the expected way. We do this by fixing the directions that correspond to all but one share to a fixed value.
If possible, we try to fix them to $0$ or some other value that allows easy descriptions of the output based on the one varying share. Then, we observe how the model outputs relate to the final share. If the hypothesis about model behavior is correct, we can also directly derive the values for a secret share from these patched outputs. Finally, after deriving secret shares, we can use Signal-to-Noise Ratio (SNR)[7] to plot where in the trace these shares leak to derive which secret share is which, i.e., which of the two shares is the mask and which the masked S-box output. Note that this patching

---

[6] For higher $d$, the structure will be in higher dimensions.

[7] SNR measures the signal variance versus the noise variance. In SCA, a higher SNR indicates a stronger signal compared to noise, making it easier to extract sensitive information.

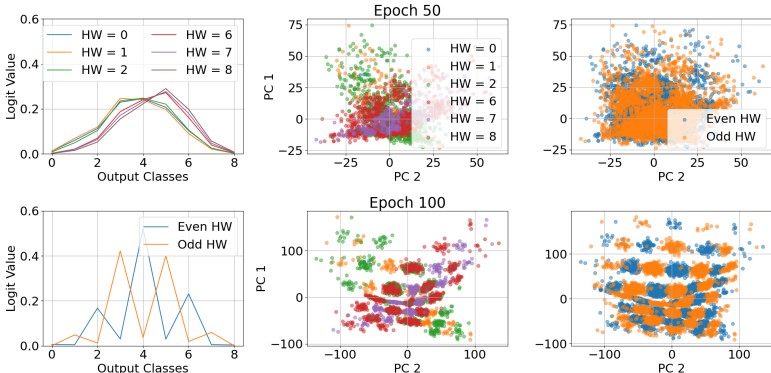

Figure 3: Logit analysis (first column) and activation analysis (remaining columns) from models at epoch 50 (top) and epoch 100 (bottom) for CHES_CTF. Legends for activation analysis are shared within columns. The difference in the number of points between the last two columns is due to not plotting the points for classes (HWs) 3, 4, and 5.

setup follows the activation patching method used in [35] without requiring a priori knowledge of secret share values.

## 4 Experimental Results

This section presents results for three common public SCA targets - CHES_CTF, ESHARD, and ASCAD (see Appendix B for details). The models are Multilayer Perceptron (MLP) neural networks with their hyperparameters taken from [35] for ESHARD and ASCAD (see Appendix C). For CHES_CTF, we directly train the ESHARD model without additional hyperparameter tuning. Note that we focus on MLP and CNN architectures as these are generally sufficient for state-of-the-art performance in SCA [34]. The analyses given here are similar (although somewhat more cumbersome) for CNNs (see Appendix G).

### 4.1 CHES_CTF Dataset

For the CHES_CTF target, we see in Figure 4 that there are two concrete increases in perceived information during training. The initial increase starts at epoch 15 and is completed around epoch 40. After another plateau in PI, there is a second increase between epochs 70-85, after which there are no more significant changes in PI.

As we aim to find what is learned during these performance jumps, we show both average logits for different classes and the two main PCs in Figure 3. After the first increase, at epoch 50, the predictions on the test set differentiate between high HW values and low HW values. When we use this information to plot PCs in the first layer (middle plot in Figure 3), we see that one diagonal corresponds to high HWs and the other to low HWs. This indicates that the HWs of both secret shares mask and masked S-box output leak in the HW leakage model and that these are the features that map onto the PCs. Further details are in Appendix H.

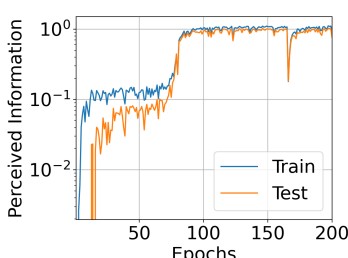

Figure 4: Evolution of Perceived Information for training and test traces of the CHES_CTF dataset.

Looking at the logits after the second performance increase at epoch 100, Figure 3 shows that in addition to the high-low HW divide, the models also separate even-odd HWs. Plotting the same components but separating even-odd HWs shows a grid structure of even and odd points. In this grid, the number of changes in even-odd is about nine, corresponding to the nine possible HW values. The even-odd separation also clearly corresponds to learning the parity of a target value from HWs

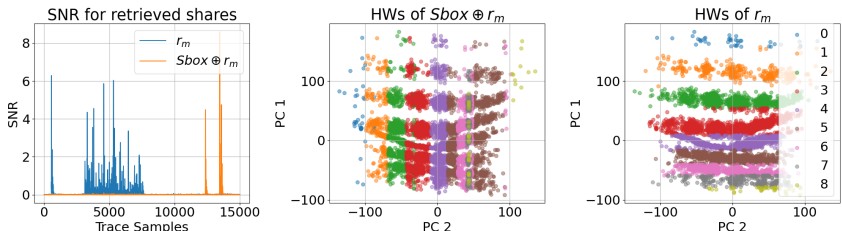

Figure 5: SNR plot and PC distributions for mask values using patching experiments for CHES_CTF. We set PC0 to -20 for both patching experiments, as that resulted in more apparent separation during manual testing.

with Boolean masking (see Appendix H.2). This leads to the ability to learn the mask values that the network uses for classification, as discussed next.

**Activation Patching:** To validate that the PC embeddings are causally related to model outputs, we can fix one of the components and observe the effects on model outputs. An additional consideration is that when we fix the value of one of the Hamming weights to 0 (or 8), the output of the model should be the HW value of the other share (or 8-output if we fix the first to 8).[8] As such, if the PCs relate to mask values, we can patch one share to 0 (or 8) to retrieve the value of the other share.

To practically extract mask values, we fix the value of one PC to be (near) one of the corners of the grid we see in Figure 3. Then, we examine the model outputs to verify whether the predicted value has changed as expected. As the model generally predicts HW values between 3-5 (because those occur most), we sort each trace by the difference of logits for high (5-8) and low (0-3) HWs. Since we know the expected number of occurrences for each HW[9], we can take the first 1/256 values to be $HW = 0$, then the next 8/256 values for $HW = 1$, and so on.

The resulting mask and masked S-box distribution are shown in Figure 5. We can see that fixing values of certain PCs to extremes results in the model basing its predictions mainly on the other PC, as is expected when one of the shares is fixed to 0 (or 8). When we visualize the SNR for each share, we observe clear spikes corresponding to the usage of the leaking values. First, we see spikes related to the value of $r_m$, indicating the loading of the mask and some pre-processing before the encryption. Then later, we see leakage related to S-box$[p_i \oplus k_i] \oplus r_m$. Due to the page limit, ESHARD results are in the Appendix E. In summary, there is only one generalization spike, which results in the ability of the model to distinguish high-low HWs. The results are qualitatively the same as for CHES_CTF.

## 4.2 ASCAD Dataset

For the ASCAD target (the main benchmark for DLSCA research since its introduction in 2018 [2]), generally the ID leakage model is used as this results in better attack performance [2, 34]. As such, for this dataset, we additionally train linear probes for each bit of both the S-box input and output.

Figure 6 shows a sudden transition to positive PI from epochs 8-12, corresponding to increased probe accuracies for the input bits. Immediately after, PI still increases marginally until improvement stops at epoch 25. This increase

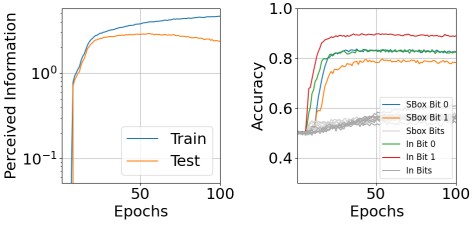

Figure 6: Evolution of PI and probe accuracies for bits during training for the ASCAD dataset.

is accompanied by the increasing probe accuracies for the two least significant S-box output bits.

---

[8]Note that patching one share to be 0 to validate that the outputs become directly related to the other share has been done before in [35] although by using knowledge of the masking randomness.

[9]If the mask values are uniformly distributed, which they generally are for the security properties to hold [20].

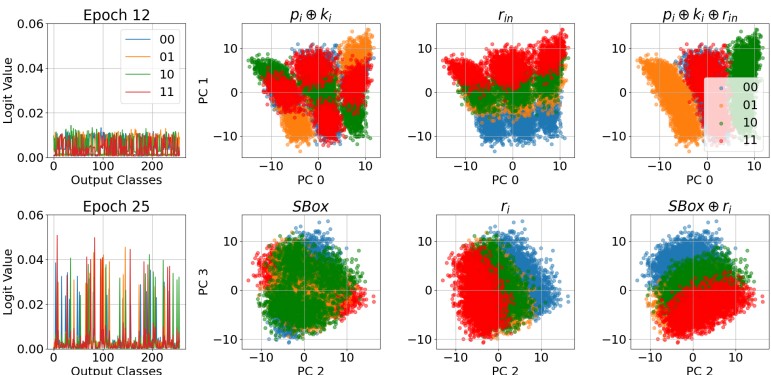

Figure 7: Logit analysis for two LSBs of $p_i \oplus k_i$ at epoch 12 and S-box$[p_i \oplus k_i]$ at epoch 25 with corresponding actual mask values for ASCAD. Note that for the lower logit plot, we use only traces with $p_i \oplus k_i$ in 00 for clarity, and that extracted mask values are in Figure 12.

Indeed, the two least significant bits (LSBs) for both input and output clearly achieve far higher accuracies than the other bits, which only marginally improve over random guessing.

Looking at the logits in Figure 7, at epoch 12, the values are distributed according to the two input bits. When we plot PCs to distinguish the values of these bits in Figure 7, we see an emerging structure in the first two PCs of the activations in the second layer corresponding to the combination of mask values by mapping these on certain axes. Note that the grid structure in both cases follows a $3 \times 3$ structure over the more ideal $4 \times 4$ if all four possible 2-bit values of the masks are perfectly distinguished. This is due to the physical leakage of two classes for the secret shares (mostly) overlapping, as shown in the two rightmost plots.

When we consider the logits at epoch 25 for the output bits,[10] the mean values are significantly higher. Additionally, the logits are spread out across fewer values. This aligns with the network's predictions, which now incorporate the information on the output bits. We also observe a visually similar structure to the grid at epoch 12 appearing in the 3rd and 4th PCs for the S-box output bits. The first two PCs remain related to the input bits as in epoch 12. Within the activation patching experiments for ASCAD, we observe causal effects on outputs by training probes on the final layer and selectively intervening on key components. However, further refinement is needed to extract mask values accurately. The experiments are presented in Appendix F.

## 5  Discussion

Recently, several works have characterized the algorithms learned by networks trained on simple algorithmic tasks, e.g., modular arithmetic and more general group operations [30, 7, 56]. Furthermore, other studies have identified individual circuits that perform grammatical operations in language models [48, 33]. These works showcase that interpretable algorithms are learned during discrete phase transitions, aligned with the neural scaling law from [28], which states that network training is a collection of discrete 'quanta' that correspond to (potentially interpretable) circuits. The eventual goal of interpreting a neural network then becomes to enumerate all of the phase transitions during training. In this work, we showcase that this type of ambitious interpretability can be possible for models trained on real-world datasets in SCA.

While this is a positive result, the broader relevance of this is somewhat limited. As mentioned in the introduction, side-channel data poses challenges due to noise and unpredictable physical leakage characteristics, but it is also very structured, and the number of relevant features is (expected to be) very low. The core task of the networks for SCA, combining secret share leakage to recover the target intermediate value, is very similar to the group operations learned in [30]. Indeed, in the networks we investigate, there are at most two jumps in performance, allowing for more detailed examination

---

[10]We fix the input bits to 00 to increase visibility, for a complete description, see Appendix I.1.

of each individual case. Additionally, the low number of expected features avoids issues that are prevalent in larger models trained on more general tasks (e.g., superposition [10]).

A key difference between our approach and those in previous works on group operations is that we assume no knowledge of input features due to our threat model. This then prohibits the use of standard input interventions, as we cannot replace input features. As such, we need to derive features from the model outputs and find which inputs are grouped together by the model. By finding these groups, we can work backwards from the model outputs to find relevant input features. This still requires domain-specific knowledge as, otherwise, it might not be immediately obvious that the grid structure in Figure 3 relates to the embedding of secret share HWs.

Another consideration is that our analysis is focused on model activations and outputs. We do not use other available information, like gradients or input visualizations, in this work. Some previous works have focused on visualizing what parts of the input the network is using (see Appendix A for some examples). However, without access to masking randomness, it is difficult to relate this to which shares and intermediate values are (not) being exploited. In contrast, the extraction of secret shares from model activations allows us to generate SNR graphs that match the visualizations in Figure 8 while splitting the individual shares. For automating analyses, gradient information could potentially be used, e.g., to identify phase transitions during training.

One of the most interesting points in our results is that the second performance jump in the CHES_CTF model seems to rely on the first. The high-low HW embeddings of the secret-share features gets progressively refined to be a direct embedding of the HW. When these embeddings are sufficiently clean, the model can distinguish even-odd HWs, as described in Appendix H.2. This indicates that adapting SCA training methodologies to first identify 'simpler' leakage and then build upon it with more complex leakage models could be a promising future direction for making models more efficient.

## 6    Relevance To DLSCA

As DLSCA becomes more common, it is increasingly important to understand how neural networks exploit implementations. This work provides concrete analyses for several common side-channel datasets, showing the possibility of reverse engineering masks from network activations. We show that specific structures can occur for different side-channel targets, indicating that building a library of common structures could be useful in analyzing future networks. This is especially relevant as masking schemes are often similar across ciphers (i.e., post-quantum ciphers often also use Boolean masking). As networks are often trained to recombine shares, these structures should be shared across different ciphers.

As our main practical result, we can reverse engineer secret shares from a trace by using the structures learned by the neural networks. To our knowledge, only [13] can extract mask values, where this work is focused on a specific implementation using classical side-channel techniques (thus, without considering machine learning approaches), which requires stronger assumptions than DL-based attacks. Extracting mask values substantially benefits evaluations, as we can move from black-box to white-box evaluations. This, in turn, would allow for better feedback to designers of cryptographic implementations.

One might question whether this is relevant for attackers, as we require a model that already breaks (one key byte of) the target. When attacks target individual bytes, the difficulty of breaking any individual byte can vary, even for the same device. As such, when masks are shared for all bytes (which is often required for masking non-linear operations, e.g., the S-box in AES), spending significant effort to break one key byte might allow retrieving the shared mask. Subsequent attacks against other key bytes become more straightforward as we can use the retrieved mask during training to effectively move the attack to an unprotected case by including the mask, see the white-box evaluations in [5].

Finally, discovering how neural networks concretely defeat countermeasures can improve evaluation/attack methodologies and countermeasure design. On the evaluation/attack side, we can design more effective methods for label distribution that consider common mistakes networks make, which can improve convergence [50]. On the defense side, understanding what type of leakage is more/less easily exploited could lead to the design of more (cost-)effective countermeasures that enable more

robust protections. Concretely, for the Boolean masking schemes we consider in this work, the structures that arise from the model embeddings of the secret share leakages into PCs naturally form the high-low structure. Further, when these embeddings are sufficiently refined, even-odd clusters emerge, indicating that using masking schemes that are less algebraically compatible with practical leakage functions, like prime-field masking [11], could also be beneficial for protecting against DLSCA.

Profiled attacks against real-world targets are often significantly more complex than idealized evaluation settings, where the same device is used for both profiling and attack. Differences between devices often result in worse performance when models trained on a profiling device are applied to the target [3]. In security evaluations, the same device is commonly used for both profiling and attack to represent the worst-case scenario where the device differences are minimal. As such, the attack sets of the considered targets are from the same device as the profiling set, which raises questions about the practical relevance of these results for real-world settings. However, this work considers post-hoc explanations for models that already break a target. Therefore, the experimental evaluations emulate what is possible even for (more realistic) non-profiled adversaries that obtain a trained model using techniques like [45], as non-profiled attacks always consider a single device.

Doing similar analyses in practice might still be difficult, especially for side-channel evaluators with limited expertise in DL. This work is then aimed at highlighting that ambitious post-hoc interpretability in DLSCA is feasible. Future work can build on these results to find more automated approaches to aid evaluators in performing these analyses in practice. Notably, automatically finding relevant features by matching novel model outputs to (variations of) features found in this work using, e.g., KL-Divergence, seems promising. Variations here can include what leakage model is expected (e.g., which bits leak, using Hamming Distance between operations over HW) and what specific operations leak (e.g., using earlier/later operations in AES).

## 7 Conclusions and Future Work

We show that interpreting neural networks trained on side-channel models is feasible, even without access to random masks. Moreover, we highlight the effectiveness of investigating the structures learned during discrete steps in model performance and find evidence for the weak universality of circuits in side-channel models. Finally, we leverage these insights to reverse engineer the mask values. Automating these analyses represents an interesting direction for future work. Additionally, further work on leveraging the insights into DLSCA models to improve evaluation methods could be useful. For example, using tailored leakage models that consider common structures could help simplify model tuning. Finally, we only focus on MLPs (and a CNN in Appendix G) as these networks provide state-of-the-art attack performance for the tested targets. Extending this to Transformer-based architectures also used in DLSCA [16, 15] is an interesting direction for future work.

### Acknowledgments

The authors would like to thank Gorka Abad, Abraham Basurto-Becerra, Azade Rezaeezade and anonymous reviewers for valuable feedback which helped improve this work. This work was (in part) supported by the Dutch Research Council (NWO) through the Challenges in Cyber Security (CiCS) project of the Gravitation research program under the grant 024.006.037, and through the PROACT project with grant number NWA.1215.18.014.

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

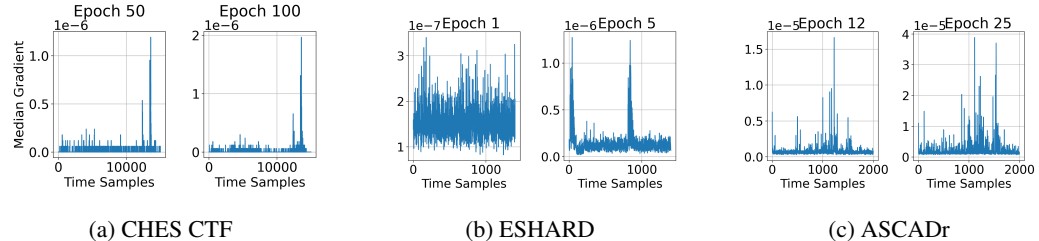

(a) CHES CTF         (b) ESHARD         (c) ASCADr

Figure 8: Gradient Visualization Example of MLP models on three datasets.

## A   Extended Related Work

The first main direction for interpreting what leakage neural networks exploit was the input visualization approach. Here, input attribution techniques from the vision domain were adopted to show regions of the traces that influence network predictions [18]. Several input attribution methods have been investigated, where examples include gradient-based visualizations in [26], weight-based visualizations in [55], and occlusion-based approaches [51]. In Figure 8 we show an example of gradient-based visualization for the MLPs used in this work. As we can see, the visualizations do highlight similar regions as we found using the secret-shares we extracted. However, from only these plots it is difficult to extract meaningful insights without access to secret-share randomness. In the works proposing these explainability approaches, visualization is often compared with ground-truth leakage information using SNR for known secret-shares (as we do using extracted shares), which can then highlight the leakage from which shares the network is exploiting [26, 18]. However, this assumes knowledge of masking randomness.

More recently, the internals of networks have also been analyzed. Van der Valk et al. [46] compared networks trained on different side-channel datasets, showing that these networks are often very different from each other. Wu et al. [49] used ablations to evaluate the roles of specific layers in defeating certain countermeasures. In Perin et al. [35], the probes are trained at each layer for several (ir)relevant secret shares, showing that an information bottleneck forms, resulting in the compression of irrelevant share information while relevant values are maintained.

Finally, there are two works that propose using more interpretable architectures for DLSCA. Yap et al. used a truth table convolutional network to find SAT-equations for important points in the trace [52]. Yoshida et al. used Kolmogorov-Arnold networks [53]. These works show nice interpretations of network behavior, but these interpretations are only shown on simulations or selected informative features. The interpretability benefits of using these architectures trained on full-length traces are still an open question, as the additional (non-informative) points result in less 'clean' interpretations. These more interpretable architectures also come at an additional computational cost, while resulting in worse attacks, limiting their relevance as practical replacements for standard MLPs and CNNs.

Overall, these works provide limited insights into how certain countermeasures are defeated. Although some approaches can show which shares are (not) exploited, these require access to masking randomness during training, which is not always possible. Our work provides an approach that allows us to interpret standard neural network architectures, while minimizing the necessary assumptions beyond a network that can break a target. Besides this, our approach is the only approach that allows for the extraction of secret shares.

## B   Datasets

We utilize publicly available datasets commonly used in SCA literature for benchmarking. These datasets implement AES-128 with Boolean masking protection. The attack set consists of 10 000 traces for each dataset.

CHES CTF 2018 [19][11] consists of power consumption measurements from an AES-128 implementation running on ARM Cortex-M4 (32 bits). CHES CTF 2018 raw traces contain 650 000 sample

---

[11]Referred to as CHES_CTF.

points per trace. Following [34], we take a subset of 150 000 points corresponding to the initial setup and the first AES round and resample to 15 000 samples per trace. The profiling set has 30 000 traces.

ESHARD-AES128 [47][12] consists of EM measurements from a software-masked AES-128 implementation running on an ARM Cortex-M4 device. The AES implementation is protected with a first-order Boolean masking scheme and shuffling of the `S-box` operations. In this work, we consider a trimmed version of the dataset that is publicly available [13] and includes the processing of the masks and all `S-box` operations in the first encryption round without shuffling. This dataset contains 100 000 measurements with 90 000 traces for the profiling set.

ASCAD [2] measures EM emissions from an AES-128 implementation on AVR RISC (8 bits). We use the version with the variable key in the profiling set. The traces are 250 000 sample points per trace. Following [35], we take a window of 20 000 points, which are resampled to 2 000 points. 200 000 traces are used for profiling.

## C   Models and Training

The used models are MLPs from [35], where model configurations were found through a random hyperparameter search for ESHARD and ASCAD. Note that, as the ESHARD model performed well directly for CHES_CTF, we did not do further optimizations.

The model for CHES_CTF and ESHARD is a 4-layer MLP with 40 neurons in each layer with *he_uniform* weight initialization. We use *relu* activations. We use the Adam optimizer with a learning rate of 0.0025 and L1 regularization set to 0.000075. The batch size is 400, and we train for 200 epochs for CHES_CTF and 100 for ESHARD.

For ASCAD, the model is a 6-layer MLP with 100 neurons in each layer with *random_uniform* weight initialization. We use the Adam optimizer with a learning rate of 0.0005. We use *elu* activations, and we again train for 100 epochs with a batch size of 400.

## D   Computational Load

Training these models takes under an hour on a desktop workstation with 64GB RAM and an NVIDIA 4080 GPU. Producing PI/probe plots per epoch takes a similar time (mainly because of reloading models for every epoch from disk). All other experiments take negligible compute (seconds, sometimes minutes).

## E   ESHARD Results

In Figure 9, we see that for ESHARD, only one phase transition occurs for the test set. At epoch 4, the perceived information becomes positive, and the models start to generalize. We note that the main distinction here is again the high-low HWs, similar to the first step in CHES_CTF. Further analyses are analogous to CHES_CTF, although the model here can never distinguish between even and odd HWs.

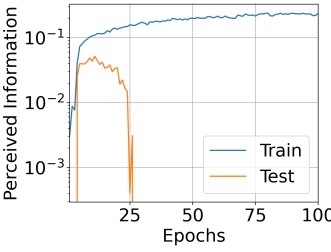

Figure 9: Evolution of Perceived Information for train and test traces of the ESHARD dataset.

---

[12]Referred to as ESHARD.

[13]https://gitlab.com/eshard/nucleo_sw_aes_masked_shuffled

In the two rightmost plots in Figure 10, we showcase distributions of the concrete intermediate values the models use. The models are clearly mapping the HWs of secret shares onto specific features.

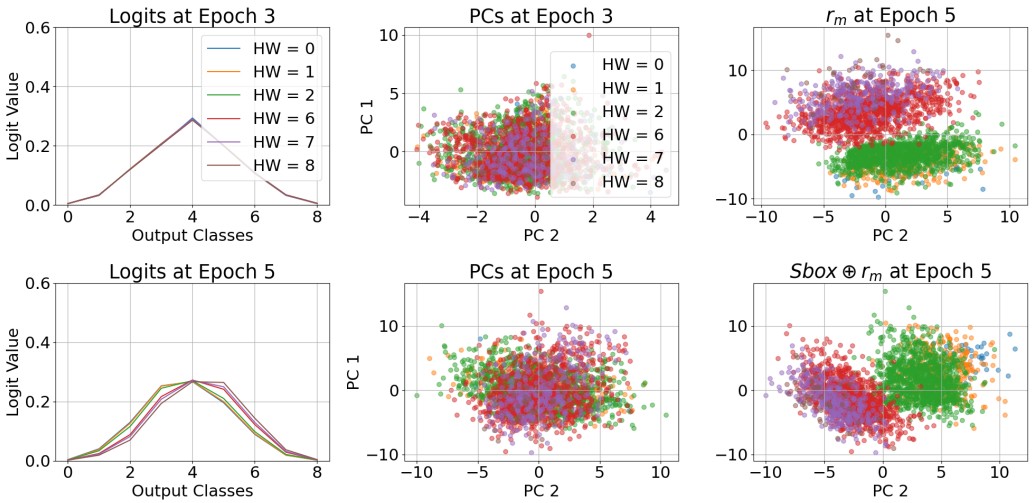

Figure 10: Logit analysis (first column) and activation analysis (second column) from models at epoch 3 (top row) and epoch 5 (bottom row). The legend is shared among all figures. We also include the PC embeddings for the actual mask of secret shares at epoch 5 (third column). The masks we extract are in Figure 11.

## E.1 Activation Patching

We can do similar patching experiments as done for CHES_CTF in Section 4.1. As the high-low HWs are not on the diagonals in the PCs at epoch 5, we rotate the PC coordinates before patching and then rotate them back before continuing inference to align PCs more with the expected masks. The results we see in Figure 11 closely match the actual distributions of secret share HWs as seen in the rightmost plots of Figure 10.

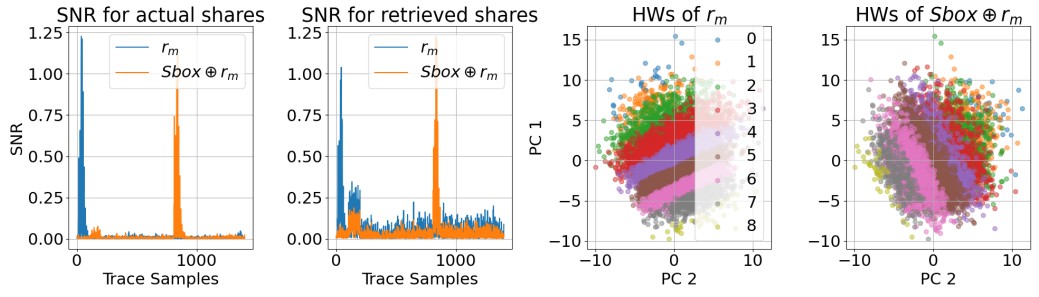

Figure 11: SNR plot and PC component distributions for mask values using patching experiments in ESHARD.

## F  ASCAD Patching results

As the leakage model for ASCAD is more complicated than the HW models, patching becomes more difficult. First, we train probes on the final layer to classify the input and output bits separately. We can directly measure the effects on only the input or the output. Patching the input shares in the PC components in layer 2, which we show in Figure 7, does not work. Then, we find a qualitatively similar structure in PC1 and PC2 in layer one and patch there.

For the patches on the output shares, we set the first two components, which are related to the input shares, to 0 to isolate the effects of the patched components. For both experiments, we again rotate the two components by multiplying them with a rotation matrix to simplify the patches.

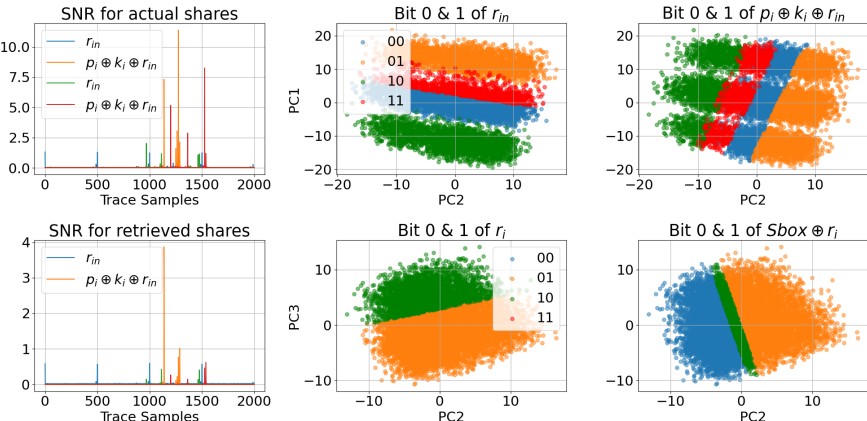

Figure 12: SNR plot and PC component distributions for mask values using patching experiments in ASCAD.

In Figure 12, we can see that the patches work reasonably well. Clearly, intervening on the found components has some causal effects. Furthermore, as we can see in the SNR plots, the patched outputs of the models are tied to the mask values we expect. However, the SNR values are significantly lower than those for the actual shares, and the $r_i$ and S-box $\oplus r_i$ shares only result in two or three classes, respectively, where we expect four. Additionally, the reversed shares do not combine to the correct label for the input bits, indicating that while the mask values we retrieve are a reasonable clustering, further post-processing is necessary to retrieve the actual values.

As we aim to keep the experiments (somewhat) aligned across all targets, we do not tailor the patching methods further for ASCAD. The current experiments show we can intervene in the structures and observe effects on the (probe) outputs. However, refining mask extraction methods in models with more complicated interactions is an interesting direction for future work. We provide further analysis to validate model predictions based on the four bits in Appendix I.1.

## G    ASCAD CNN

To see whether analyses are feasible for CNNs, we consider the CNN used for the ASCAD target from [35]. In Figure 13, we see that the bits that show significantly increased accuracies are the same

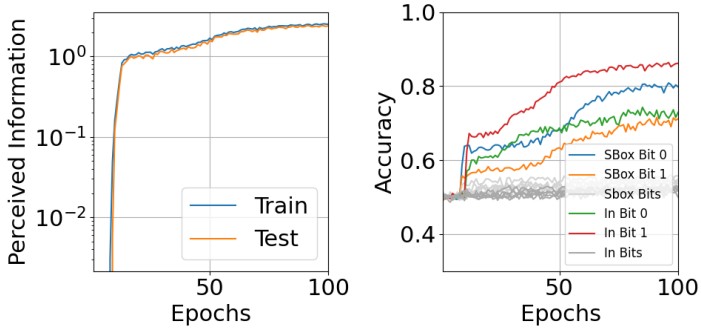

Figure 13: Evolution of Perceived Information and probe accuracies for bits during training for the ASCAD dataset for CNN.

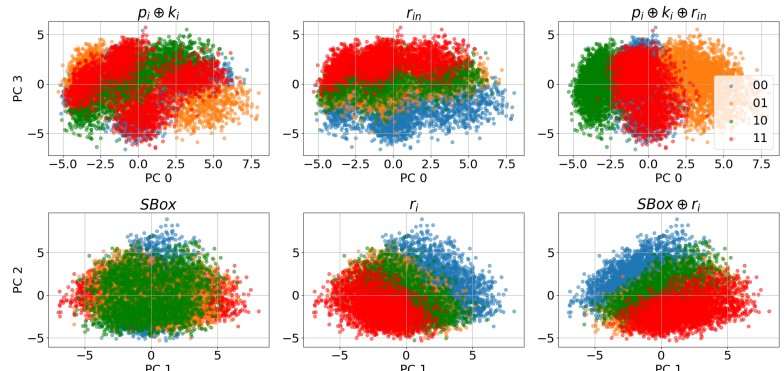

Figure 14: PC structures for ASCAD CNN after the third convolutional block.

as in Figure 6. However, the performance increase (after initial, smaller jumps) is more gradual. This, and the model having significantly more layers to check, results in it being somewhat more difficult to find the structures.

As the performance increases steadily, we focus our analysis on the final model at epoch 100 and show these in Figure 14. We note that in this case, shares are still in the first 4 PCs, but the input shares are not the first 2 PCs, but PC 0 and 2 (output shares PC 1 and 3).

## H  HW Recombination CHES_CTF and ESHARD

Next, we discuss how mask recombination can be done algorithmically for the HW leakage model.

### H.1  High-Low HW Distinguishing

For the CHES_CTF and ESHARD targets, we notice that after the first performance increase (for some cases), high and low HWs can be differentiated. These are byte-based implementations protected with Boolean masking with order 2, i.e., the sensitive value $x = x_1 \oplus x_2$ ($\oplus$ being bitwise xor). When, based on prior experience working with these targets, we then choose to model the leakage (and therefore the presumed features of the model) as the $L = HW(x_i)$[14] we can consider modeling how occurrences of different classes $Y = HW(x)$ look. In Table 1, low HWs tend to be on the diagonal from top-left to bottom-right, while high HWs tend to be on the other diagonal. This (low HWs on one diagonal while high HWs are on the other) matches the PC embeddings for both models in Figure 10 and Figure 3.

### H.2  Even-Odd HW Distinguishing CHES_CTF

For CHES_CTF, we further see that the even and odd HW target classes can be distinguished after the second performance jump. From Table 1, it is clear that if the HWs of each secret share can be retrieved accurately enough, there should be a clear separation between even and odd HWs for the resulting point. Indeed, for any point $HW(x_1), HW(x_2)$ where $x = x_1 \oplus x_2$ we have that $HW(x_1) + HW(x_2) \bmod 2 = HW(x) \bmod 2$. This can be seen in Table 1 for two 8-bit shares, but the ability to distinguish the parity of $HW(x)$ holds for general higher masking orders $d$ [21].

## I  Bitwise Leakage ASCAD

As we show in Figure 6, the features the model learns for ASCAD are the two least significant bits of both $p_i \oplus k_i$ and S-box$[p_i \oplus k_i]$. We first note that the way the first two bits of S-box$[p_i \oplus k_i]$ relate to model labels (S-box$[p_i \oplus k_i]$) is straightforward: if bits 0 and 1 of S-box$[p_i \oplus k_i]$ are 00,

---

[14]We note that we knew this a priori for ESHARD and it was strongly suspected for CHES_CTF. However, it is also a common leakage model in practice.

| $HW(x)$ | Matrices counting occurrences of $HW(s_1), HW(s_2)$ s.t. $x = s_1 \oplus s_2$ from 0-9. |
|---|---|
| HW = 0 
 HW = 1 | $$\begin{bmatrix} 1 & 8 & 0 & 0 & 0 & 0 & 0 & 0 & 0 \\ 8 & 8 & 56 & 0 & 0 & 0 & 0 & 0 & 0 \\ 0 & 56 & 28 & 168 & 0 & 0 & 0 & 0 & 0 \\ 0 & 0 & 168 & 56 & 280 & 0 & 0 & 0 & 0 \\ 0 & 0 & 0 & 280 & 70 & 280 & 0 & 0 & 0 \\ 0 & 0 & 0 & 0 & 280 & 56 & 168 & 0 & 0 \\ 0 & 0 & 0 & 0 & 0 & 168 & 28 & 56 & 0 \\ 0 & 0 & 0 & 0 & 0 & 0 & 56 & 8 & 8 \\ 0 & 0 & 0 & 0 & 0 & 0 & 0 & 8 & 1 \end{bmatrix}$$ |
| HW = 2 
 HW = 3 | $$\begin{bmatrix} 0 & 0 & 28 & 56 & 0 & 0 & 0 & 0 & 0 \\ 0 & 56 & 168 & 168 & 280 & 0 & 0 & 0 & 0 \\ 28 & 168 & 336 & 840 & 420 & 560 & 0 & 0 & 0 \\ 56 & 168 & 840 & 840 & 1680 & 560 & 560 & 0 & 0 \\ 0 & 280 & 420 & 1680 & 1120 & 1680 & 420 & 280 & 0 \\ 0 & 0 & 560 & 560 & 1680 & 840 & 840 & 168 & 56 \\ 0 & 0 & 0 & 560 & 420 & 840 & 336 & 168 & 28 \\ 0 & 0 & 0 & 0 & 280 & 168 & 168 & 56 & 0 \\ 0 & 0 & 0 & 0 & 0 & 56 & 28 & 0 & 0 \end{bmatrix}$$ |
| HW = 4 | $$\begin{bmatrix} 0 & 0 & 0 & 0 & 70 & 0 & 0 & 0 & 0 \\ 0 & 0 & 0 & 280 & 0 & 280 & 0 & 0 & 0 \\ 0 & 0 & 420 & 0 & 1120 & 0 & 420 & 0 & 0 \\ 0 & 280 & 0 & 1680 & 0 & 1680 & 0 & 280 & 0 \\ 70 & 0 & 1120 & 0 & 2520 & 0 & 1120 & 0 & 70 \\ 0 & 280 & 0 & 1680 & 0 & 1680 & 0 & 280 & 0 \\ 0 & 0 & 420 & 0 & 1120 & 0 & 420 & 0 & 0 \\ 0 & 0 & 0 & 280 & 0 & 280 & 0 & 0 & 0 \\ 0 & 0 & 0 & 0 & 70 & 0 & 0 & 0 & 0 \end{bmatrix}$$ |
| HW = 5 
 HW = 6 | $$\begin{bmatrix} 0 & 0 & 0 & 0 & 0 & 56 & 28 & 0 & 0 \\ 0 & 0 & 0 & 0 & 280 & 168 & 168 & 56 & 0 \\ 0 & 0 & 0 & 560 & 420 & 840 & 336 & 168 & 28 \\ 0 & 0 & 560 & 560 & 1680 & 840 & 840 & 168 & 56 \\ 0 & 280 & 420 & 1680 & 1120 & 1680 & 420 & 280 & 0 \\ 56 & 168 & 840 & 840 & 1680 & 560 & 560 & 0 & 0 \\ 28 & 168 & 336 & 840 & 420 & 560 & 0 & 0 & 0 \\ 0 & 56 & 168 & 168 & 280 & 0 & 0 & 0 & 0 \\ 0 & 0 & 28 & 56 & 0 & 0 & 0 & 0 & 0 \end{bmatrix}$$ |
| HW = 7 
 HW = 8 | $$\begin{bmatrix} 0 & 0 & 0 & 0 & 0 & 0 & 0 & 8 & 1 \\ 0 & 0 & 0 & 0 & 0 & 0 & 56 & 8 & 8 \\ 0 & 0 & 0 & 0 & 0 & 168 & 28 & 56 & 0 \\ 0 & 0 & 0 & 0 & 280 & 56 & 168 & 0 & 0 \\ 0 & 0 & 0 & 280 & 70 & 280 & 0 & 0 & 0 \\ 0 & 0 & 168 & 56 & 280 & 0 & 0 & 0 & 0 \\ 0 & 56 & 28 & 168 & 0 & 0 & 0 & 0 & 0 \\ 8 & 8 & 56 & 0 & 0 & 0 & 0 & 0 & 0 \\ 1 & 8 & 0 & 0 & 0 & 0 & 0 & 0 & 0 \end{bmatrix}$$ |

Table 1: Occurrences of HWs for two 8-bit shares for each of the nine output classes, cell $i, j$ in each matrix corresponds to $HW(s_1), HW(s_2)$. For the percentage of examples in practice, these values should be divided by $256^2$. As even and odd $HW(x)$ never occur in the same place, we show two HWs in one matrix. Note that any red value (resp. black) is a zero in black (resp. red).

then these correspond to predicting each label $y \bmod 4 \equiv 0$. For bits 0 and 1 of $p_i \oplus k_i$, we can use the inverse of the S-box[15]. If we define $y' = \text{S-Box}^{-1}[y]$ then if bit 0 and 1 of $p_i \oplus k_i$ are 00, we predict $y$ s.t. $y' \bmod 4 \equiv 0$.

Combining these, we can divide the output classes into 16 clusters corresponding to model predictions. Practically, we define the outputs that belong to the 16 classes as $Y_i = \{y | y \equiv i \bmod 4 \wedge y' \equiv \lfloor \frac{i}{4} \rfloor \bmod 4\}$. Here, we set $i$ to be a concatenation of bit 1 and 0 of $p_i \oplus k_i$ and then bit 1 and 0 of $\text{S-box}[p_i \oplus k_i]$.

---

[15]The AES S-box is bijective, which simplifies this, but the analysis also works for surjective functions by taking the pre-image.

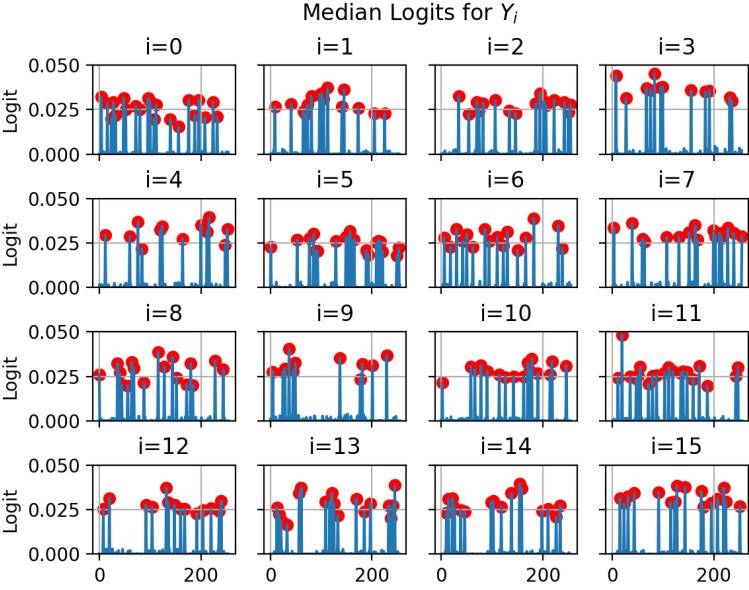

Figure 15: Median logits for traces belonging to varying $Y_i$ classes. The red dots indicate indices in the respective $Y_i$.

We can then train a linear probe on the activations of the final layer to predict these 16 classes. If we then transform the probe outputs to evenly distribute the predictions for its $i$'th output to the values in $Y_i$, we can measure the entropy between this resulting distribution and the model outputs. In summary, the probe accuracy is 0.64, and the PI between the probes' transformed distribution and the labels is 2.47 vs. 2.58 for the actual model. The entropy (in bits) between the probe outputs and model predictions is 0.27, indicating that most of the relevant behavior is explained by using the probe.

## I.1 Logits For ASCAD with Classes

In Figure 15, we show the median logits for traces belonging to classes $Y_i$. As we can see, the logits corresponding to the expected points in $Y_i$ are always the main peaks.

Figure 16 shows how logits change from epoch 12 to epoch 25. When we analyze using only S-box inputs, we see that the logit values are significantly higher before the accuracies for output bits are increased. This is explained by the fact that each of these cases combines four plots (vertically) in Figure 15. Concretely, as for each trace in Figure 16, we combine traces that belong to 4 different classes of the output bits, we expect the logits for each index that belongs to $p \oplus k \mod 4 \equiv i$ to only be high for 1/4 traces, resulting in lower medians. Note that mean values do not show this same trend, as the increase in the $Y_i$ class compensated for this decrease.

To verify that the results in Figure 15 are not an artifact of selecting traces, we visualize the same analysis for bits 2 and 3 in Figure 17. Clearly, the output values are significantly lower than for correct bits, indicating that these bits are (mostly) not being used by the model.

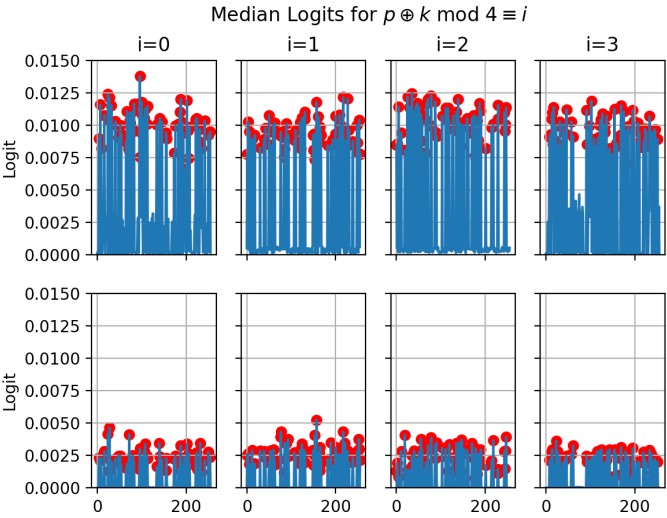

Figure 16: Median logits for traces belonging to varying classes of 2 LSBs of S-box input for epoch 12 (top) and 25 (bottom).

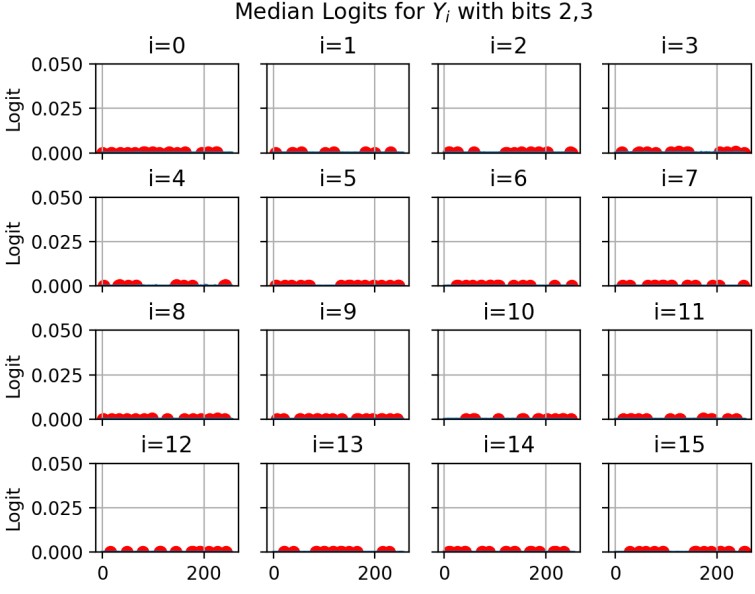

Figure 17: Median logits for traces belonging to varying $Y_i$ for bits 2 and 3. The red dots indicate indices in the respective $Y_i$.

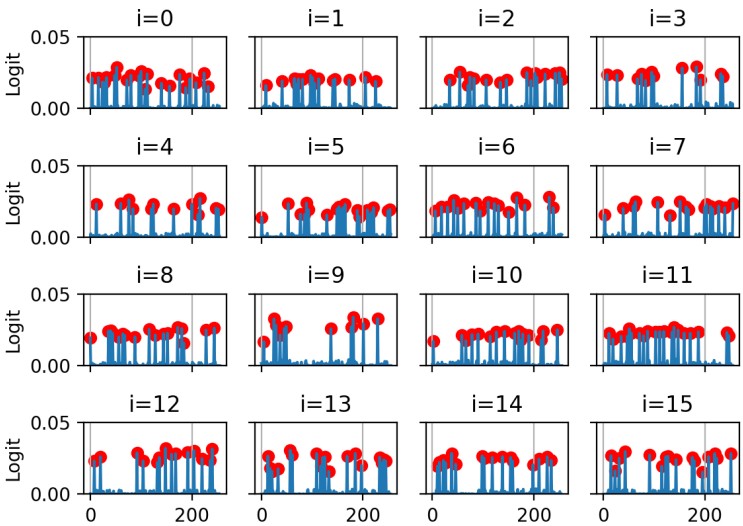

Figure 18: Median logits for traces belonging to varying $Y_i$ classes for CNN model. The red dots indicate indices in the respective $Y_i$.

