# OpenReview forum: "Interpreting Emergent Features in Deep Learning-based Side-channel Analysis"
_NeurIPS.cc/2025/Conference — NeurIPS 2025 poster_

### Official Review · Reviewer_CmhV · 2025-06-23

**Clarity:** 3
**Significance:** 3
**Originality:** 3
**Rating:** 5
**Confidence:** 4

**Summary:**

This paper applies techniques from mechanistic interpretability (MI) to studying what features deep learning (DL) model exploit in side channel analysis (SCA). It examines features in models trained on three different implementations of AES 128 and uses these to trace back to causal model outputs. The result shows that there is some alignment among features learned by different models.

**Questions:**

- This is minor, but the title does not match the content of the paper very well. Why not simply call it "Interpreting Features in Deep Learning-based Side-channel Analysis"? It wasn't clear from the paper why the phrase "you have to be realistic" was included in the title.

- Line 328: "We show that specific structures can occur for different side-channel targets, indicating that building a library of common structures can be useful in analyzing future networks." The current work studies only SCA of AES128, which is a relatively straightforward algorithm. Do you think the methods that you develop here can apply to SCA of other cryptosystems (like CRYSTALS-KYBER https://eprint.iacr.org/2023/1084)?

**Ethical Concerns:**

["NO or VERY MINOR ethics concerns only"]

**Final Justification:**

The rebuttal adequately addressed my concerns. My score remains the same.

**Limitations:**

- The authors could have given more consideration to how features discovered via MI could be leveraged for defense against SCA attacks. Why was this question not explored?

**Quality:**

3

**Strengths And Weaknesses:**

Strengths:
- Investigation of an important and interesting problem: what features make DL models so successful at SCA?
- Creative use of MI tools to study this problem.
- Evaluation of techniques across several implementations of AES 128 and model types, indicating repeatable results.

Weaknesses:
- Although universal features are identified across the SCA models evaluated, no defense is proposed (although discussion suggests interpretability techniques could be helpful for designing defenses).
- Only evaluates a specific cryptosystem (AES 128), leaving open the question of whether these techniques work on other, perhaps more complicated, systems.
- Only one MI technique (activation analysis) is tested. It would be interesting to explore if other MI techniques provide different insights.
- Some paragraphs are *very* long and wander, making multiple points. Tightening up the writing and creating paragraphs that address a specific point would improve paper quality.

---

> ### Author Rebuttal · Authors · 2025-07-30
>
> W1: The analyzed results are mostly aimed at showcasing how to do post-hoc analysis of neural nets that break a target. In our view, understanding how networks exploit leakage is a prerequisite for designing efficient, targeted countermeasures. We can provide more discussion on different masking schemes that avoid the structures present in this work, which (intuitively) may be more difficult for NNs to learn (e.g., prime-field masking [CMM+23]). However, as there are (almost) no publicly available targets (and datasets) that implement such masking schemes, this will remain somewhat speculative and is an interesting direction for future work.
>
> W2Q2: We focus on AES targets as those are common in the related literature, and these provide representative benchmarks of masked implementations. Moreover, AES is a complex target compared to many newer algorithms, e.g., lightweight crypto. The concepts behind the attacks on post-quantum crypto (like Kyber) and symmetric crypto (like AES) are rather different, so one would need to adapt leakage models and consider different operations that leak.
>
> We also note that how models extract physical leakage and combine secret shares should (broadly) be the same for targets masked with similar (Boolean) schemes. The models are being used to extract and recombine leakage from secret shares to improve predictions for a target value. As such, we expect that mask recombination should look similar for similar masking schemes across ciphers. We can extend the discussion on this.
>
> W3: We emphasize that analyzing activations (and activation patching) are very general approaches where the specifics we use in this paper are only one option (which works well for these cases). It is an interesting direction for future work to use/adapt some more automated techniques from recent MI literature on LLMs, although we note those approaches often assume some knowledge of what input features are relevant a priori, which corresponds to knowledge of masking randomness in SCA. This is not always possible in practice, even in evaluation settings (see [23]).
>
> W4: We will shorten paragraphs and improve the writing.
>
> Q1: The `You have to be realistic' was a reference to extending the work on toy models on group operations to real-world ML problems. We can remove it if it causes confusion.
>
> [CMM+23] Cassiers, G., Masure, L., Momin, C., Moos, T., & Standaert, F.-X. (2023). Prime-Field Masking in Hardware and its Soundness against Low-Noise SCA Attacks. IACR Transactions on Cryptographic Hardware and Embedded Systems, 2023(2), 482-518. https://doi.org/10.46586/tches.v2023.i2.482-518

---

> > ### Comment · Reviewer_CmhV · 2025-08-01
> > **Response**
> >
> > Thanks for your comments. My score remains the same, though I would appreciate it if the promised changes were made to the paper.

---

### Official Review · Reviewer_LCay · 2025-06-24

**Clarity:** 2
**Significance:** 3
**Originality:** 3
**Rating:** 4
**Confidence:** 3

**Summary:**

This paper uses mechanistic interpretability to understand how deep learning models extract secrets in side-channel attacks. It reveals what leakages models rely on, recovers secret masks, and shows that such interpretability is feasible even in noisy, real-world settings, making security evaluation more transparent.

**Questions:**

1. Why are only logits and activations chosen for analysis in this work? A clear motivating experiment is needed to show that only logits and activations are enough to analyse to understand the neural networks trained for SCA.

2. It is unclear whether the proposed analysis is effective compared to prior work. A comparison table is required to understand the effectiveness of the proposed analysis.

3. Why is the analysis shown only for one model [31]? To better understand the effectiveness , the proposed analysis should be done on atleast one more model (For eg: model from [32])

**Ethical Concerns:**

["NO or VERY MINOR ethics concerns only"]

**Final Justification:**

Most of my concerns are addressed in the rebuttal and I have increased my score accordingly. However, it is still not clear from Sec.4 why only logits and activations are selected for the analysis. Maybe an analysis/experiment that show that logits and activation are the only features that in a network can be used for this analysis and other features are not relevant.

**Limitations:**

Yes

**Paper Formatting Concerns:**

No major formatting issues

**Quality:**

2

**Strengths And Weaknesses:**

Strengths:

* The paper works on an important problem of the interpretability of neural networks trained for SCA.

* By applying mechanistic interpretability to deep learning-based SCA models, the work makes an important and novel contribution in bridging the gap between high-performing yet opaque models and the need for transparency in security evaluations.

* The authors make a compelling case for transitioning from black-box performance benchmarking to white-box understanding of model behavior. This shift is valuable for designing more robust defenses.

* The proposed analysis is evaluated on diverse datasets.

Weaknesses:

* The paper analyses the logits and activations of the neural network models trained for SCA. It is unclear why only these two components are chosen for analysis. A detailed justification/motivation for the same is required.

* It is important to compare with prior works, such as methods that focus on input visualization [22,14] and masking randomness [48,31]. This will establish the effectiveness of the proposed analysis.

* The proposed analysis is evaluated using only one model [31]. It is important to do the same analysis on other models trained for SCA, such as [32].

* While the paper addresses an important topic, it lacks a clear novel theoretical or methodological contribution. The analysis primarily builds on known observations (e.g., [49]) and does not introduce new techniques or frameworks, which limits its overall impact.

* The code and pretrained models do not appear to be available at the time of submission. Consequently, it is not currently feasible to verify the reproducibility of the reported results. Providing access to the implementation, along with detailed setup and usage instructions, would substantially improve the transparency, accessibility, and impact of this work for the broader research community.

---

> ### Author Rebuttal · Authors · 2025-07-30
>
> W1Q1: `Why are only logits and activations chosen for analysis in this work? A clear motivating experiment is needed to show that only logits and activations are enough to analyze and understand the neural networks trained for SCA.'
> We showcase several examples in Section 4 where we can understand what features networks are learning during sharp performance increases. This then enables us to retrieve the secret shares from network activations using activation patching, a first in DLSCA. Overall, we believe that the experiments in Section 4 clearly demonstrate the effectiveness of the approach for understanding (and validating) network behavior.
>
> Besides this, we note that analyzing activations is standard in mechanistic interpretability, as the goal of our work is to understand what is happening inside the models (i.e., what the model is representing in the activations). MI techniques that work with inputs are not possible, as we do not know a priori what features are present in the input.
>
> W2Q2: As the methods for interpretability are qualitative and make different assumptions and/or have different goals, it is difficult to directly compare performance in a table. Input visualization only focuses on inputs and does not split the input attribution to individual secret shares (which we can achieve with SNRs for retrieved shares in Figure 5 (left)). Methods using masking randomness are not comparable, as we do not assume to know this randomness (which is common in practice, see [23]). We can add extended related work with some examples of input visualization to the appendix to provide a more elaborate discussion.
>
> W3Q3: We evaluate using 3 different models from [31] (obtained using random search, which is standard for these targets, see [32]) to improve upon the interpretability from [31] with fewer assumptions (i.e., unknown masking randomness) (CNN in appendix). We note that running searches for model architectures/hyperparameters for new targets is relatively standard in DLSCA. Adding more models from different works is possible, but as the models we use provide state-of-the-art performance for the current targets, other models will (presumably) learn the same features ([32] is a SoK paper and does not contain specific models).
>
> W4 limited novelty: We note that the application of MI techniques to different domains is very limited. In addition, as we do not assume knowledge of input features (i.e., masking randomness), we need to construct features from output logits, which requires adaptation of MI techniques. Furthermore, the setup that we provide for analyzing networks in DLSCA using mechanistic interpretability, the connection of the `initial plateau' [23] to feature emergence as in [26,49], and the application of activation patching in intermediate layers to retrieve masks are novel.
>
> W5: We provide the link to an anonymous repository with code, extracted datasets (and scripts to generate those from raw measurements from original repositories), and pretrained model checkpoints right after summarizing our contributions and before the background section, page 3, line 80-81). We can move the link to the abstract so that it is visible on the first page if that would help.

---

> > ### Comment · Reviewer_LCay · 2025-08-04
> > **Response**
> >
> > I would like to thank the authors for their detailed response. Most of my concerns are addressed and I have increased my score accordingly. However, it is still not clear from Sec.4 why only logits and activations are selected for the analysis. Maybe an analysis/experiment that show that logits and activation are the only features that in a network can be used for this analysis and other features are not relevant.

---

> > > ### Author Response · Authors · 2025-08-04
> > > **RE: Response**
> > >
> > > Thanks for your engagement in the review process. We are happy to have cleared up most of the concerns.
> > >
> > >
> > > Why we only look at logits and activations:
> > >
> > > Overall, we emphasize that there are broadly three categories that we can analyze to understand the forward pass of models from an MI perspective:
> > >  - Inputs: In our case, these are side-channel traces. The applicability of methods for mechanistic understanding here is impossible, as controlling inputs is necessary, which is impossible for masked implementations without known masking randomness. By our patching experiments and the extraction of relevant masking randomness, we can tie model predictions back to leakage of specific values in specific locations in the traces.
> > >  - Model internals: activations, for analyzing what the model is representing internally, and how these intermediate representations affect the model outputs.
> > >  - Outputs: model outputs, i.e., logits
> > >
> > > As controlling inputs is infeasible without knowledge of masking randomness, we focus on outputs and model activations. In our view, the experiments showcase that the features we find with our method describe model behavior well. We can include some additional discussion on this in Section 4.
> > >
> > >
> > > For less trace-specific analyses, it could be possible to utilize model gradients during the feature emergence (or differences in model weights before/after) to also understand what the model has learned during these phases, or potentially even detect when the model is learning new features more automatically. However, (to the best of our knowledge), no other methods provide concrete tools to do this and extract information about specific input examples. Therefore, we consider it out of scope, but this seems like an interesting direction for future work, and we can include a short discussion of this in the conclusion.

---

> > > > ### Author Response · Authors · 2025-08-07
> > > >
> > > > As the discussion phase is closing soon, we were wondering whether this answer for why we consider the analysis of logits and activations is clear? If you have any remaining questions/clarifications, we would be happy to discuss those.

---

### Official Review · Reviewer_8DGR · 2025-07-09

**Clarity:** 2
**Significance:** 2
**Originality:** 2
**Rating:** 4
**Confidence:** 2

**Summary:**

This paper is about Side-channel analysis (SCA) for understanding deep learning (DL) networks. The paper elaborates that SCA can help to understand network learning and can be used to attack the DL networks. The paper used 4-layers MLP network (160 neuron) for the experiments and used Hamming Weight and changes in it to measure the understanding and activities in the network. The experiments used two datasets that then networks' perceived information and learning (accuracy) is measured and related to HW. The discussion mentioned that it is not results are not conclusive but their implication show that there is a possibility of proposed idea.

**Questions:**

Please look into Strengths and weaknesses mentioned above.

**Ethical Concerns:**

["NO or VERY MINOR ethics concerns only"]

**Final Justification:**

Authors have replied to most of the limitations in the rebuttal. However, if authors embedded all the revisions mentioned in the rebuttal, then it will be a significant change to the original manuscript. I still think that authors need to look into more recent networks other than MLP and CNN.

**Limitations:**

Experiments are limited. They need to clearly support the suggested contributions.

**Paper Formatting Concerns:**

Paper is not well written and there are issues regarding references which needs to be addressed

**Quality:**

1

**Strengths And Weaknesses:**

strengths
- Idea is interesting and novel.
Weaknesses.
- article is not well-written.
- References are not available using the DOIS such as [19],[20],[22],[24] (serious issue)
- Many references include reports or opinions or archives which not published in peer-reviewed venues such as [4], [27],[37]. This makes the introduction weak.
- Introduction as well as background are not clear about the contribution. On one hand, it is mentioning that SCA is used as a vulnerability of the DL architectures while on the other hand, it is the reason for explainability. The introduction needs to have clarity so readers can be led to clear contributions
-  The abbreviation are used before explaining them such as HW
- The justification of SCA is not clear too as there are no studies clearly mentioning that it can be used well.
- Profiling attack require copy of the network. If network's copy is available then why SCA is needed as entire network can be studied and interpreted easily.
- It is not clear how physical characteristics of the network will be accessed to deployed DL network
- 4-layers MLP is not an optimal selection as state of the art network for experiments. Existing networks are far more complicated like transformers or MAMBA and it is not clear how these results can be extended to recent networks
- On one hand Discussion clearly mentions that SCA is not conclusive on MLP while on the other hand it is projected as giving inside network information. This shows that there is not clear contribution of the study.
- Experimental results do not clearly support the contributions
- discussion is mostly about implications rather than the results

---

> ### Author Rebuttal · Authors · 2025-07-30
>
> In your summary, the claim is that our paper uses SCA for understanding DL networks, and here we want to emphasize that this is not the case. We use mechanistic interpretability methods and propose a procedure to conduct explainability analysis that aims to explain what DL networks learn when used to perform deep learning-based SCA (DLSCA) on cryptographic implementations. We are interested in understanding how the side-channel leakage of a running crypto system is exploited by the neural network to extract the secret information (key in this case).
> The recent emergence of DLSCA has resulted in many papers (see [32]) and is becoming standardized for evaluations of cryptographic implementations required for certification (see [9]). Understanding DLSCA and what the models actually learn is therefore an important open question.
> We discuss the main issues below.
>
> Minor issues:
> - We will add DOIs for the works where these are available. Thanks for pointing this out.
> - Abbreviations:  HW is Hamming weight. We will double-check all abbreviations. Thanks for pointing this out.
>
> Weaknesses:
>  - Article is not well written: We will clarify long paragraphs as pointed out by reviewer CmhV. If you have any further actionable concerns about the writing, we are happy to address them.
>  - Non-peer-reviewed references in introduction: transformer-circuits is where Anthropic posts many papers ([4,8,27] have 100s of citations) and sometimes provides independent reviews. [37] is meant to show real-world vulnerabilities found by evaluation labs in industry (which are not commonly reported in scientific publications). Other references are preprints or textbooks [40]. We acknowledge that this is not ideal, but as mechanistic interpretability and DLSCA are relatively new fields, this is unavoidable.
>  - `Introduction as well as background are not clear about the contribution. On one hand, it is mentioning that SCA is used as a vulnerability of the DL architectures while on the other hand, it is the reason for explainability. The introduction needs to have clarity so readers can be led to clear contributions': We do not mention SCA is used as a vulnerability of DL architectures in the paper, as we are focused on cryptographic implementations. The explainability of DLSCA is a key aspect of performing evaluations (see [9,32]).
>  - SCA cannot be used well: This seems to say we do not justify the use of SCA for understanding DL networks, so again, we clarify that this is not what we are doing. We use mechanistic interpretability to understand what networks are learning in the DLSCA use case on crypto systems. However, we argue that we do justify why we investigate explainability for DLSCA. As we discussed in the introduction, deep learning-based SCA has emerged as (one of) the main tools to perform SCA evaluations in industry labs and is becoming standardized [9] (in addition to real-world attacks [37,38]). In performing the evaluations, explainability of DLSCA is a key aspect (see [9,32]).
>  - Profiling attacks are a well-established class of attacks for evaluating cryptographic implementations under worst-case assumptions. Profiling SCA requires a clone device running the same cryptographic implementation, not a copy of a neural network.
>  - Architectures were taken directly from [31]. Commonly, simple MLPs and CNNs work well in DLSCA; therefore, our work considers relevant networks for DLSCA (note that similar architectures achieve optimal attacks in related works).  Extending the work to other architectures is an interesting direction for future work.
>  - `On one hand, the Discussion clearly mentions that SCA is not conclusive on MLP while on the other hand, it is projected as giving inside network information. This shows that there is no clear contribution of the study.': We are not sure what this means. We would be grateful for further clarification on the reviewer's point here.
>  - Experimental results do not clearly support the contributions: We respectfully disagree; we provide experiments with various MLPs and a CNN. We can extract masking randomness, indicating that we provide a better understanding of what leakage models exploit in several cases.
>  - Discussion is mostly about implications rather than the results: We can expand the discussion on results, which, for now, is mostly contained in Section 4.

---

> > ### Author Response · Authors · 2025-08-05
> >
> > Thank you again for taking the time to review our paper. As the discussion period comes to a close, we want to make sure we've addressed your main concerns. Are there any remaining questions or points you'd like us to clarify?

---

> > ### Comment · Reviewer_8DGR · 2025-08-06
> >
> > Thanks to the authors for submitting the rebuttal.
> > I would like to clarify that that missing DOIs and unreviewed reference provided in the review was example of many instances. This means that authors need to go over all the references. The major problem with the article is that it is not well-written.
> >
> > Authors have shared their commitments to improve the article as per suggestions and limitations provided in the review. If all these suggestions are embedded then article will have to go through significant changes (pretty much re-writing entire article).

---

> > > ### Author Response · Authors · 2025-08-06
> > >
> > > We would like to thank the reviewer for their response.
> > >
> > > In our view, the changes we propose to the weaknesses the reviewer states are relatively minor one/two-sentence clarifications (which, respectfully, were generally already contained in the paper and missed by the reviewer). We will fix DOIs and abbreviations, and tighten up the writing of long (and winding) paragraphs as mentioned by reviewer CmhV.
> > >
> > > To be precise on the clarifications already in the paper:
> > >
> > >   - SCA against cryptography, not DL-models: We mention AES, keys, plaintexts throughout the paper. We are still unsure where the reviewer got the idea that we use SCA as a vulnerability of the DL architectures.
> > >
> > >   - Justification SCA: lines 24-28 in intro discuss both evaluation uses and real-world attacks using SCA.
> > >
> > >   - How profiled attacks are used: abstract and introduction discuss evaluations, lines 92-109 give more BG on profiled attacks
> > >
> > >   - Architectures: We mention from where we take architectures, and that MLPs are generally sufficient in DLSCA in lines 216-222. We can add a short paragraph on this second point in the discussion to clarify further (see our rebuttal W3Q3 for reviewer LCay)
> > >
> > >   - `On one hand, the Discussion clearly mentions that SCA is not conclusive on MLP while on the other hand, it is projected as giving inside network information. This shows that there is no clear contribution of the study.'We state our concrete contributions clearly on lines 66-79. We are still unsure what the reviewer means here and would appreciate further clarification on this question.
> > >
> > >   - Discussion on results:  We are still not quite sure what discussion the reviewer is missing specifically, as we feel we have covered the results in Section 4 (and appendices with additional results). We acknowledge that Section 5 is mostly about broader implications, but we think it's important to discuss these in the main paper. Due to page limits, a lot of additional results (and discussion of those) are in the appendices.

---

### Note · Authors · 2025-08-12

Dear AC and reviewers,

We sincerely thank all reviewers for their time, constructive feedback, and engagement during the rebuttal phase.

During the rebuttal, we mainly clarified some misunderstandings and answered questions about potential future directions. Reviewers LCay and Cmhv are happy with our clarifications and answers to their questions, which we will also include in the text where necessary. Additionally, we will update the manuscript to provide an extended related work with examples of input visualization techniques as suggested by reviewer LCay, and work to shorten some of the long paragraphs as suggested by reviewer CmhV.

Reviewer 8DGR points out some missing DOIs in references, which we already fixed. Besides this, reviewer 8DGR points out some misunderstandings that may be possible when reading the paper and indicates that our clarifications would result in significant changes. In response, we point to where in the original paper those misunderstandings are already addressed. We believe our comments address all raised issues but did not receive further feedback from the reviewer.


Sincerely,
The Authors

---

### Decision · Program_Chairs · 2025-09-17

**Decision:**

Accept (poster)

**Comment:**

The paper investigates how neural networks succeed at deep learning-based side-channel analysis (SCA). The authors apply mechanistic interpretability to models trained on cryptographic tasks. They observe that models learn in discrete jumps. These performance gains correspond to the emergence of clear, geometric structures within the network's hidden activations. From a machine learning perspective, the network spontaneously learns a disentangled representation of the hidden causal factors (secret shares) from extremely noisy, high-dimensional input. The authors use this insight to reverse-engineer the model's learned algorithm and successfully recover secret cryptographic masks.

Strengths. The paper applies mechanistic interpretability to a complex, real-world security problem. It provides a compelling case study of a simple MLP learning a multi-step algorithm from scratch. This work demonstrates how a network can discover and separate hidden variables. It is quite impressive that one could causally intervene on the system and recover secret values.

Weaknesses: presentation and limited scope. One reviewer raised significant concerns about the clarity of the writing and the quality of the references, with which AC agrees. The experimental validation is also constrained to the AES cryptosystem and relatively simple model architectures, which raises questions about generalisability.

Rebuttal & discussion. 8DGR criticised the writing quality. Others put more weight on the novelty and significance, trusting the authors that they'll revise the paper appropriately. The final consensus was that the paper's technical contribution is solid and important, provided that the authors undertake a significant revision to improve clarity. **AC strongly urges and requires that the paper go through the promised revision to improve the writing and presentation before the camera-ready.**

Why accept? The decision to accept is based on the novelty and significance of the findings. The paper offers a rare demonstration of successful reverse-engineering of a neural network's learning dynamics. The contribution is valuable both to the security and ML interpretability communities.